# Mechanism of Producing Metallic Nanoparticles, with an Emphasis on Silver and Gold Nanoparticles, Using Bottom-Up Methods

**DOI:** 10.3390/molecules26102968

**Published:** 2021-05-17

**Authors:** Basil Raju Karimadom, Haya Kornweitz

**Affiliations:** Chemical Sciences Department, Ariel University, Ariel 4077625, Israel; basilr@ariel.ac.il

**Keywords:** nanoparticles, silver nanoparticles, gold nanoparticles, DFT, standard reduction potentials

## Abstract

Bottom-up nanoparticle (NP) formation is assumed to begin with the reduction of the precursor metallic ions to form zero-valent atoms. Studies in which this assumption was made are reviewed. The standard reduction potential for the formation of aqueous metallic atoms—E^0^(M^n+^_aq_/M^0^_aq_)—is significantly lower than the usual standard reduction potential for reducing metallic ions M^n+^ in aqueous solution to a metal in solid state. E^0^(M^n+^_aq_/M^0^_solid_). E^0^(M^n+^_aq_/M^0^_aq_) values are negative for many typical metals, including Ag and Au, for which E^0^(M^n+^_aq_/M^0^_solid_) is positive. Therefore, many common moderate reduction agents that do not have significantly high negative reduction standard potentials (e.g., hydrogen, carbon monoxide, citrate, hydroxylamine, formaldehyde, ascorbate, squartic acid, and BH_4_^−^), and cannot reduce the metallic cations to zero-valent atoms, indicating that the mechanism of NP production should be reconsidered. Both AgNP and AuNP formations were found to be multi-step processes that begin with the formation of clusters constructed from a skeleton of M^+^-M^+^ (M = Ag or Au) bonds that is followed by the reduction of a cation M^+^ in the cluster to M^0^, to form M_n_^0^ via the formation of NPs. The plausibility of M^+^-M^+^ formation is reviewed. Studies that suggest a revised mechanism for the formation of AgNPs and AuNPs are also reviewed.

## 1. Introduction

Nanoparticles (NP) constitute a special class in chemistry that is situated between molecules and crystals. As such, they possess unique properties that open new scientific horizons. Indeed, the physical [1,2], biological [3,4], and chemical [5,6] characteristics of gold and silver nanoparticles (AuNPs and AgNPs, respectively) confer on them wide applicability [7] in medicine [8,9,10,11,12,13,14,15,16,17,18,19,20,21], agriculture [22,23,24,25,26], photonics [12,18,20,27,28,29,30], microelectronics [31,32,33], and catalysis [30,34,35,36,37,38,39]. The plasmonic properties of Au and Ag thus stimulate a wide range of research in nanoscale optics, photonics and sensors [40]. Owing to the varied and unique properties exhibited by NPs, nanoparticle research is interdisciplinary, involving scientists from the fields of chemistry, biology, engineering, biotechnology, and material sciences.

The synthesis of metallic nanoparticles in general, and specifically the production of gold and silver nanoparticles, is very simple experimentally. Accordingly, numerous methods have been developed to derive these nanoparticles that exploit their unique properties [7,41,42,43,44,45,46,47]. The subject of fruitful, ongoing research [7,45], the different NP production methods are based on chemical, physical, and biological approaches, and can be classified as top-down or bottom-up [48] according to the initial reactants. To control the morphology, particle size, and structure of the produced NPs, template methods [49,50] that employ soft or hard templates are used in a process that also influences the NPs’ unique physical and chemical properties. Recently, a novel physical procedure for the production of Au quantum dots at room temperature was reported by Li et al. [51]. Other physical methods, such as photoreduction, are used to synthesize these NPs [52,53,54]. This review focuses on bottom-up methods, wherein the synthesis comprises nucleation and growth steps. In contrast to the wealth of experimental research on nanoparticle production, scant attention has been given to the study of its mechanisms. When considering bottom-up methods, the metallic ions are often assumed to be reduced to generate metallic atoms, and the aggregation of this is thought to form the nanoparticles. This key assumption, however, is often found to be in contradiction with the standard reduction potentials.

From 2000 to 2020, according to SciFinder, of the 852,987 papers that were published on NPs, 23% (197,080 papers) were on Au or Ag NPs. Though density-functional theory (DFT) is an important tool for the study of any process mechanism at the molecular level, only 2562 of these papers (1.3% of the publications on AuNPs or AgNPs) used DFT calculations, a number that represents only about 0.3% of all publications on NPs (Figure 1). Molecular dynamics is used only after the nascent NPs are formed to calculate their growth to their final size and shape [55,56,57,58,59,60,61]. In addition to original research, about 51,000 reviews of the work on NPs were published, of which 6612 (13%) were also on AuNPs or AgNPs, but only 41 of these include DFT calculations (Figure 2). Insofar as DFT calculations are the most common method to theoretically explore a process, the limited number of publications that exploited DFT indicates that the theoretical study of the production of AgNPs and AuNPs has received limited attention. 

Only a few publications refer to the reduction mechanism of the metallic cations and the formation of nascent NPs, and they emphasize the role of either the stabilization agent (solvent or another reactant) [62,63,64,65,66,67,68] or the capping agent [69,70]. Pilay et al. studied the nucleation mechanism of AgNPs and AuNPs on a rutile surface, with the role of vacancies studied by calculating adsorption energies [71]. The interaction between Au_9_ and Ag_9_ (a cluster of nine atoms) with pentanal in DMSO was calculated by Saldías et al. [67], while Tamuly et al. [72] calculated the reduction of Ag^+^ using pedicellamide, describing the nucleation and growth process of AgNPs. Though many publications refer to the reduction of Au^III^→Au^I^ [66,69,73], Barngrover et al. demonstrated the reduction mechanism of Au^III^→Au^I^→Au^0^ [64]. While Graham et al. [74] noted the role of ion–ion aggregation, the formation of dimers of AuX_2_^−^ are essential in the mechanism suggested by Barngrover et al. [64]. In addition, Hudgens et al. [75] demonstrated agglomeration prior to reduction. Regarding Brust–Schiffrin synthesis [76], the early accepted assumption about this method by Schaaff et al. [77] has been that polymeric Au^I^ thiolate [Au^I^SR]n can be generated as intermediate precursors [77,78,79,80,81]. Precursor species of these reactions have been identified by Zhu et al. [82]. In this review, we summarize the generally assumed mechanism of the production of nascent AgNPs and AuNPs (the nucleation step) in a bottom-up process—from a single ion to a NP—and the suggested revised mechanism. As we refer only to the early first steps, which proceed from aqueous ions to small metallic clusters that aggregate to form nanoparticles, the review does not refer to NP morphology or shape.

## 2. Production of AgNPs and AuNPs—Assumed Mechanism

The formation of NPs in a bottom-up procedure is assumed to be comprised of the following steps:

M^n^L_m_ + reducing agent → M^0^_atom_ + mL + oxidized reducing agent
kM^0^_atom_ → M_k_^0^ → → M^0^-NP

In numerous examples in the literature, the metallic ions are assumed to be reduced to give metallic atoms that subsequently aggregate to form the nanoparticles; a few such examples are described below.

Auradha et al. [83] showed that the first step of gold nanoparticle production is the reduction of the gold ions to the zero-valent metal and then the agglomeration of the atoms to form nanoparticles enveloped by stabilizing biomolecules (Figure 3b). For silver nanoparticle production, Lee and Jun [43] showed that the process begins with the reduction of Ag^+^ to Ag^0^ atoms (Figure 3a). Based on the equation of De Freitas et al. [41], the reduction of AuCl_4_^−^ using citrate in an aqueous solution,
Figure 3Chemical production of NPs: (**a**) chemical synthesis method. Reprint of Figure 1C [43]. (**b**) Mechanism of Gold-NP formation. Reprinted with permission from [83]. Copyright 2015 Elsevier. (**c**) Schematic illustration for the deduced process of gold nanoparticle formation. Reprinted with permission from [84]. Copyright 2010 ACS publication.
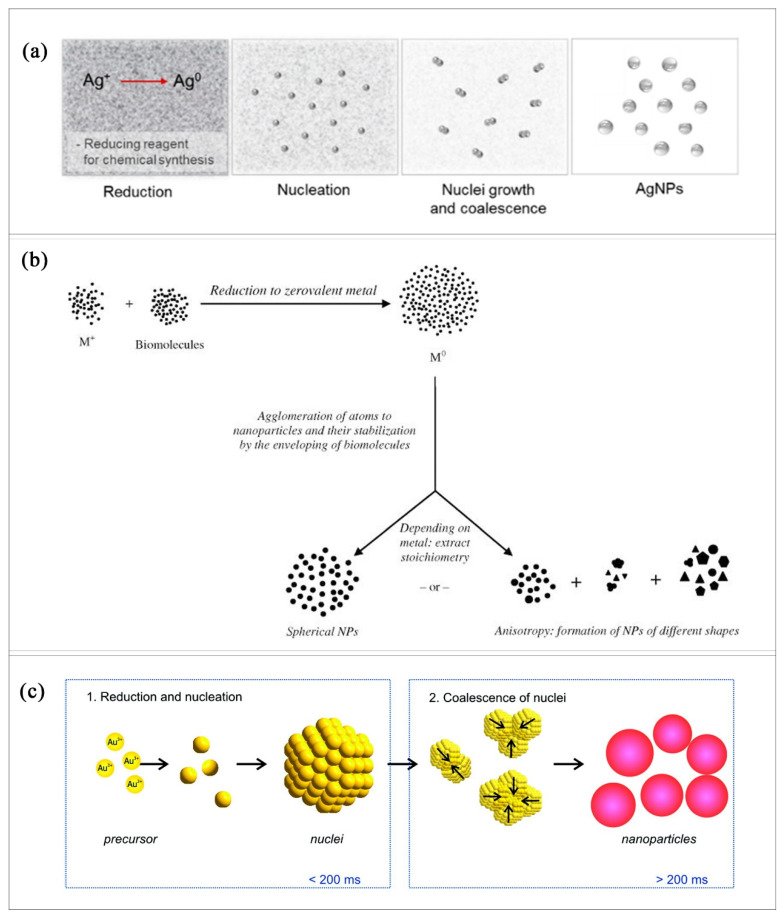

6AuCl_4_^−^ + C_6_H_8_O_7_ + 5H_2_O → 6CO_2_ + 24Cl^−^ + 6Au^0^ + 18H^+^
is one of the most common ways to synthesize gold nanoparticles. Pastoriza-Santos and Liz-Marzan [85] proposed that the reduction of silver ions to silver nanoparticles by N,Ndimethylformamide (DMF) in water takes place as follows:HCONMe_2_ + 2Ag^+^ + H_2_O ⇌ 2Ag^0^ + Me_2_NCOOH + 2H^+^

Polte et al. [84] showed (Figure 3c) that the first step of the synthesis of AuNPs is the reduction of Au^III^ to atoms and the aggregation of those atoms followed by the coalescence of the nuclei to form AuNPs. The reduction of Au^III^ to form Au^0^ is also suggested by Onesto et al. [86]. The formation of AgNPs was described by Zavras et al. [87] via the reduction of AgBH_4_ to Ag. 

Chen et al. [88] even cited the known standard reduction potential in the production of AuNPs: AuCl_4_^−^ + 2e^−^ → AuCl_2_^−^ + 2Cl^−^ E^0^ = 0.926V
AuCl_2_^−^ + e^−^ → Au^0^ +2Cl^−^ E^0^ = 1.154V

The positive reduction potentials of Au^III^ and Au^I^ given above, and of Ag^+^/Ag (0.799 V [89]), have convinced the scientific community that the first step in AuNP or AgNP formation is simply a reduction process that produces hydrated gold or silver atoms, respectively. A positive potential is required to obtain a spontaneous reaction. The positive reduction potentials given for the metallic ions are the potentials of a half reaction (reduction). Insofar as they are positive, for any common reducing agent that has a moderate non-positive reduction potential, the standard potential of the redox reaction—which comprises a reduction half-reaction and an oxidation half-reaction—will be positive and the reduction will be spontaneous. This is the case when the reduced metal is deposited on a solid surface, e.g., coating a metallic layer or an electrode, as in electroless deposition, wherein the obtained metal is in the solid state. Below are some examples of how metallic silver and gold are produced by using H_2_(g), CO(g) or BH_4_^−^(aq) as a reduction agent (the standard reduction potentials of CO(g) and BH_4_^−^(aq) are from reference [90]). The cell potential, (E^0^_cell_), is the sum of E^0^_red_, the reduction potential of the half-reaction, and E^0^_ox_, the oxidation potential of another half-reaction:

Reduction of Ag_(aq)_^+^: 

H_2(g)_ as a reducing agent:$${\mathrm{Ag}}_{\left(\mathrm{aq}\right)}^{+}+{\;e}^{-}⇆{\;\mathrm{Ag}}_{\left(s\right)}^{0}{\;\;\;\;\;\;E}_{\mathrm{red}}^{0}=0.80V$$
$$\frac{1}{2}H_{2\;\left(g\right)}⇆{\;H}_{\left(\mathrm{aq}\right)}^{+}+e^{-}{\;\;\;\;\;\;E}_{\mathrm{ox}}^{0}=0.00V$$
$${\mathrm{Ag}}_{\left(\mathrm{aq}\right)}^{+}+\frac{1}{2}H_{2\left(g\right)}⇆H_{\left(\mathrm{aq}\right)}^{+}+{\mathrm{Ag}}_{\left(s\right)}^{0}{\;\;\;\;\;\;E}_{\mathrm{cell}}^{0}=0.80V$$

CO_(g)_ as a reducing agent: $$2{\mathrm{Ag}}_{\left(\mathrm{aq}\right)}^{+}+2e^{-}⇆\;2{\mathrm{Ag}}_{\left(s\right)}^{0}{\;\;\;\;\;\;E}_{\mathrm{red}}^{0}=0.80V$$
$$H_{2}O_{\left(l\right)}+{\mathrm{CO}}_{\left(g\right)}⇆{\;\mathrm{CO}}_{2\left(g\right)}+2H_{\left(\mathrm{aq}\right)}^{+}+2e^{-}{\;\;\;\;\;\;E}_{\mathrm{ox}}^{0}=0.10V$$
$$2{\mathrm{Ag}}_{\left(\mathrm{aq}\right)}^{+}+H_{2}O_{\left(l\right)}+{\mathrm{CO}}_{\left(g\right)}⇆{\mathrm{CO}}_{2\left(g\right)}+2H_{\left(\mathrm{aq}\right)}^{+}+2{\mathrm{Ag}}_{\left(s\right)}^{0}{\;\;\;\;\;\;E}_{\mathrm{cell}}^{0}=0.90V$$

BH_4(aq)_^−^ as a reducing agent:$$4{\mathrm{Ag}}_{\left(\mathrm{aq}\right)}^{+}+4e^{-}⇆\;4{\mathrm{Ag}}_{\left(s\right)}^{0}{\;\;\;\;\;\;E}_{\mathrm{red}}^{0}=0.80V$$
$$4{\mathrm{OH}}_{\left(\mathrm{aq}\right)}^{-}+{\mathrm{BH}}_{4\left(\mathrm{aq}\right)}^{-}⇆\;B{\left(\mathrm{OH}\right)}_{4\left(\mathrm{aq}\right)}^{-}+2H_{2\left(\mathrm{aq}\right)}+4e^{-}{\;\;\;\;\;\;E}_{\mathrm{ox}}^{0}=1.24V$$
$$4{\mathrm{Ag}}_{\left(\mathrm{aq}\right)}^{+}+4{\mathrm{OH}}_{\left(\mathrm{aq}\right)}^{-}+{\mathrm{BH}}_{4\left(\mathrm{aq}\right)}^{-}⇆B{\left(\mathrm{OH}\right)}_{4\left(\mathrm{aq}\right)}^{-}+2H_{2\left(g\right)}+4{\mathrm{Ag}}_{\left(s\right)}^{0}{\;\;\;\;\;\;E}_{\mathrm{cell}}^{0}=2.04V$$

Reduction of Au_(aq)_^+^: 

H_2(g)_ as a reducing agent:$${\mathrm{Ag}}_{\left(\mathrm{aq}\right)}^{+}+{\;e}^{-}⇆{\;\mathrm{Ag}}_{\left(s\right)}^{0}{\;\;\;\;\;\;E}_{\mathrm{red}}^{0}=0.80V$$
$$\frac{1}{2}H_{2\;\left(g\right)}⇆{\;H}_{\left(\mathrm{aq}\right)}^{+}+e^{-}{\;\;\;\;\;\;E}_{\mathrm{ox}}^{0}=0.00V$$
$$2{\mathrm{AuCl}}_{2\left(\mathrm{aq}\right)}^{+}+\frac{1}{2}H_{2\left(g\right)}⇆H_{\left(\mathrm{aq}\right)}^{+}+{\mathrm{Au}}_{\left(s\right)}^{0}+2{\mathrm{Cl}}_{\left(\mathrm{aq}\right)}^{-}{\;\;\;\;\;\;E}_{\mathrm{cell}}^{0}=1.15V$$

CO_(g)_ as a reducing agent: $$2{\mathrm{AuCl}}_{2\left(\mathrm{aq}\right)}^{+}+2e^{-}⇆2{\mathrm{Au}}_{\left(s\right)}^{0}+4{\mathrm{Cl}}_{\left(\mathrm{aq}\right)}^{-}{\;\;\;\;\;\;E}_{\mathrm{red}}^{0}=1.15V$$
$$H_{2}O_{\left(l\right)}+{\mathrm{CO}}_{\left(g\right)}⇆{\;\mathrm{CO}}_{2\left(g\right)}+2H_{\left(\mathrm{aq}\right)}^{+}+2e^{-}{\;\;\;\;\;\;E}_{\mathrm{ox}}^{0}=0.10V$$
$$2{\mathrm{AuCl}}_{2\left(\mathrm{aq}\right)}^{+}+H_{2}O_{\left(l\right)}+{\mathrm{CO}}_{\left(g\right)}⇆{\mathrm{CO}}_{2\left(g\right)}+2H_{\left(\mathrm{aq}\right)}^{+}+{\mathrm{Au}}_{\left(s\right)}^{0}+2{\mathrm{Cl}}_{\left(\mathrm{aq}\right)}^{-}{\;\;\;\;\;\;E}_{\mathrm{cell}}^{0}=1.25V$$

BH_4(aq)_^−^ as a reducing agent:$$4{\mathrm{AuCl}}_{2\left(\mathrm{aq}\right)}^{+}+4e^{-}⇆4{\mathrm{Au}}_{\left(s\right)}^{0}+8{\mathrm{Cl}}_{\left(\mathrm{aq}\right)}^{-}{\;\;\;\;\;\;E}_{\mathrm{red}}^{0}=1.15V$$
$$4{\mathrm{OH}}_{\left(\mathrm{aq}\right)}^{-}+{\mathrm{BH}}_{4\left(\mathrm{aq}\right)}^{-}⇆\;B{\left(\mathrm{OH}\right)}_{4\left(\mathrm{aq}\right)}^{-}+2H_{2\left(\mathrm{aq}\right)}+4e^{-}{\;\;\;\;\;\;E}_{\mathrm{ox}}^{0}=1.24V$$
$$4{\mathrm{AuCl}}_{2\left(\mathrm{aq}\right)}^{+}+4{\mathrm{OH}}_{\left(\mathrm{aq}\right)}^{-}+{\mathrm{BH}}_{4\left(\mathrm{aq}\right)}^{-}⇆B{\left(\mathrm{OH}\right)}_{4\left(\mathrm{aq}\right)}^{-}+2H_{2\left(g\right)}+4{\mathrm{Au}}_{\left(s\right)}^{0}+8{\mathrm{Cl}}_{\left(\mathrm{aq}\right)}^{-}{\;\;\;\;\;\;E}_{\mathrm{cell}}^{0}=2.39V$$

## 3. Standard Reduction Potentials

As a matter of fact, the positive reduction potentials of Au^III^, Au^I^ and Ag^+^ are the standard reduction potentials of the couples M^n+^_aq_/M^0^_solid_, while the standard reduction potentials of the relevant couples M^n+^_aq_/M^0^_aq_ are different, as the solid metal should be evaporated to form hydrated atoms, a process in which energy should be invested. The calculation of E^0^(M^n+^_aq_/M^0^_aq_) follows the equation:E^0^(M^n+^_aq_/M^0^_aq_) = E^0^(M^n+^_aq_/M^0^_solid_) − (ΔG^0^_evaporation_/nF)

Reference [91] (*n*—number of electrons involved in the redox reaction, F—faraday constant).

The value of E^0^(M^n+^_aq_/M^0^_aq_) is significantly lower than that of E^0^(M^n+^_aq_/M^0^_solid_). Henglein’s group [92,93] was the first to calculate the value of the standard reduction potential of silver: E^0^(Ag_aq_^+^/Ag^0^_aq_) = −1.8 V versus standard hydrogen electrode (SHE). While Henglein’s calculation used enthalpies, Mondal et al. [91] recalculated this value using free energy values to obtain −1.74 V versus SHE. The standard reduction potential for gold is: E^0^ (AuCl_2_^−^_(aq)_/Au^0^_aq_ +2Cl^−^_(aq)_) = −2.23 V versus SHE [91]. Negative standard reduction potentials for E^0^(M^n+^_aq_/M^0^_aq_) were also calculated for several typical metals [91]. Table 1 summarizes the values that are given by Mondal et al. [91] relative to those of E^0^(M^n+^_aq_/M^0^_solid_) [89]. The low values of E^0^(M^n+^_aq_/M^0^_aq_) in this table indicate that it is impossible to obtain solvated zero-valent atoms in an aqueous solution, wherein only reduction to a metallic solid is possible. According to these standard reduction potentials, the production of nanoparticles of these metals, e.g., AuNPs and AgNPs, using common moderate reduction agents that do not have significantly high negative reduction standard potentials—e.g., hydrogen [94,95,96], carbon monoxide [97,98], citrate [95,99,100,101], hydroxylamine [95], formaldehyde [95], ascorbate [95], squartic acid [95], and BH_4_^−^ [102,103,104,105,106,107,108,109,110,111,112]—cannot start with the simple reduction of the cations to atoms, as commonly assumed. Hafez et al. [113] experimentally confirmed the absence of such cathodic reduction behaviors in a neutral system. The release of single hydrated aqueous atoms of silver or gold without any stabilizing agent by using H_2_(g), CO(g) or BH_4_^−^(aq), as in the above example, is thermodynamically impossible, as the potential of the cell becomes negative. 

Reduction of Ag_(aq)_^+^: 

H_2(g)_ as a reducing agent:$${\mathrm{Ag}}_{\left(\mathrm{aq}\right)}^{+}+{\;e}^{-}⇆{\;\mathrm{Ag}}_{\left(\mathrm{aq}\right)}^{0}{\;\;\;\;\;\;E}_{\mathrm{red}}^{0}=-1.74V$$
$$\frac{1}{2}H_{2\;\left(g\right)}⇆{\;H}_{\left(\mathrm{aq}\right)}^{+}+e^{-}{\;\;\;\;\;\;E}_{\mathrm{ox}}^{0}=0.00V$$
$${\mathrm{Ag}}_{\left(\mathrm{aq}\right)}^{+}+\frac{1}{2}H_{2\left(g\right)}⇆H_{\left(\mathrm{aq}\right)}^{+}+{\mathrm{Ag}}_{\left(\mathrm{aq}\right)}^{0}{\;\;\;\;\;\;E}_{\mathrm{cell}}^{0}=-1.74V$$

CO_(g)_ as a reducing agent: $$2{\mathrm{Ag}}_{\left(\mathrm{aq}\right)}^{+}+2e^{-}⇆\;2{\mathrm{Ag}}_{\left(\mathrm{aq}\right)}^{0}{\;\;\;\;\;\;E}_{\mathrm{red}}^{0}=-1.74V$$
$$H_{2}O_{\left(l\right)}+{\mathrm{CO}}_{\left(g\right)}⇆{\;\mathrm{CO}}_{2\left(g\right)}+2H_{\left(\mathrm{aq}\right)}^{+}+2e^{-}{\;\;\;\;\;\;E}_{\mathrm{ox}}^{0}=0.10V$$
$$2{\mathrm{Ag}}_{\left(\mathrm{aq}\right)}^{+}+H_{2}O_{\left(l\right)}+{\mathrm{CO}}_{\left(g\right)}⇆{\mathrm{CO}}_{2\left(g\right)}+2H_{\left(\mathrm{aq}\right)}^{+}+2{\mathrm{Ag}}_{\left(\mathrm{aq}\right)}^{0}{\;\;\;\;\;\;E}_{\mathrm{cell}}^{0}=-1.64V$$

BH_4(aq)_^−^ as a reducing agent:$$4{\mathrm{Ag}}_{\left(\mathrm{aq}\right)}^{+}+4e^{-}⇆\;4{\mathrm{Ag}}_{\left(\mathrm{aq}\right)}^{0}{\;\;\;\;\;\;E}_{\mathrm{red}}^{0}=-1.74V$$
$$4{\mathrm{OH}}_{\left(\mathrm{aq}\right)}^{-}+{\mathrm{BH}}_{4\left(\mathrm{aq}\right)}^{-}⇆\;B{\left(\mathrm{OH}\right)}_{4\left(\mathrm{aq}\right)}^{-}+2H_{2\left(\mathrm{aq}\right)}+4e^{-}{\;\;\;\;\;\;E}_{\mathrm{ox}}^{0}=1.24V$$
$$4{\mathrm{Ag}}_{\left(\mathrm{aq}\right)}^{+}+4{\mathrm{OH}}_{\left(\mathrm{aq}\right)}^{-}+{\mathrm{BH}}_{4\left(\mathrm{aq}\right)}^{-}⇆B{\left(\mathrm{OH}\right)}_{4\left(\mathrm{aq}\right)}^{-}+2H_{2\left(g\right)}+4{\mathrm{Ag}}_{\left(\mathrm{aq}\right)}^{0}{\;\;\;\;\;\;E}_{\mathrm{cell}}^{0}=-0.50V$$

Reduction of Au_(aq)_^+^: 

H_2(g)_ as a reducing agent:$${\mathrm{AuCl}}_{2\left(\mathrm{aq}\right)}^{+}+{\;e}^{-}⇆{\;\mathrm{Au}}_{\left(\mathrm{aq}\right)}^{0}+2{\mathrm{Cl}}_{\left(\mathrm{aq}\right)}^{-}{\;\;\;\;\;\;E}_{\mathrm{red}}^{0}=-2.23V$$
$$\frac{1}{2}H_{2\;\left(g\right)}⇆{\;H}_{\left(\mathrm{aq}\right)}^{+}+e^{-}{\;\;\;\;\;\;E}_{\mathrm{ox}}^{0}=0.00V$$
$$2{\mathrm{AuCl}}_{2\left(\mathrm{aq}\right)}^{+}+\frac{1}{2}H_{2\left(g\right)}⇆H_{\left(\mathrm{aq}\right)}^{+}+{\mathrm{Au}}_{\left(\mathrm{aq}\right)}^{0}+2{\mathrm{Cl}}_{\left(\mathrm{aq}\right)}^{-}{\;\;\;\;\;\;E}_{\mathrm{cell}}^{0}=-2.23V$$

CO_(g)_ as a reducing agent: $$2{\mathrm{AuCl}}_{2\left(\mathrm{aq}\right)}^{+}+2e^{-}⇆2{\mathrm{Au}}_{\left(\mathrm{aq}\right)}^{0}+4{\mathrm{Cl}}_{\left(\mathrm{aq}\right)}^{-}{\;\;\;\;\;\;E}_{\mathrm{red}}^{0}=-2.23V$$
$$H_{2}O_{\left(l\right)}+{\mathrm{CO}}_{\left(g\right)}⇆{\;\mathrm{CO}}_{2\left(g\right)}+2H_{\left(\mathrm{aq}\right)}^{+}+2e^{-}{\;\;\;\;\;\;E}_{\mathrm{ox}}^{0}=0.10V$$
$$2{\mathrm{AuCl}}_{2\left(\mathrm{aq}\right)}^{+}+H_{2}O_{\left(l\right)}+{\mathrm{CO}}_{\left(g\right)}⇆{\mathrm{CO}}_{2\left(g\right)}+2H_{\left(\mathrm{aq}\right)}^{+}+2{\mathrm{Au}}_{\left(\mathrm{aq}\right)}^{0}+4{\mathrm{Cl}}_{\left(\mathrm{aq}\right)}^{-}{\;\;\;\;\;\;E}_{\mathrm{cell}}^{0}=-2.13V$$

BH_4(aq)_^−^ as a reducing agent:$$4{\mathrm{AuCl}}_{2\left(\mathrm{aq}\right)}^{+}+4e^{-}⇆4{\mathrm{Au}}_{\left(\mathrm{aq}\right)}^{0}+8{\mathrm{Cl}}_{\left(\mathrm{aq}\right)}^{-}{\;\;\;\;\;\;E}_{\mathrm{red}}^{0}=-2.23V$$
$$4{\mathrm{OH}}_{\left(\mathrm{aq}\right)}^{-}+{\mathrm{BH}}_{4\left(\mathrm{aq}\right)}^{-}⇆\;B{\left(\mathrm{OH}\right)}_{4\left(\mathrm{aq}\right)}^{-}+2H_{2\left(\mathrm{aq}\right)}+4e^{-}{\;\;\;\;\;\;E}_{\mathrm{ox}}^{0}=1.24V$$
$$4{\mathrm{AuCl}}_{2\left(\mathrm{aq}\right)}^{+}+4{\mathrm{OH}}_{\left(\mathrm{aq}\right)}^{-}+{\mathrm{BH}}_{4\left(\mathrm{aq}\right)}^{-}⇆B{\left(\mathrm{OH}\right)}_{4\left(\mathrm{aq}\right)}^{-}+2H_{2\left(g\right)}+4{\mathrm{Au}}_{\left(\mathrm{aq}\right)}^{0}+8{\mathrm{Cl}}_{\left(\mathrm{aq}\right)}^{-}{\;\;\;\;\;\;E}_{\mathrm{cell}}^{0}=-0.99V$$

## 4. Formation of M-M M-M^I^ and M^I^-M^I^ Bonds

The standard reduction potential E^0^(M^n+^/(M^0^_aq_) is significantly lower than the values of E^0^(M^n+^/(M^0^_solid_) for all of the metals, and these values are negative for many common metallic ions. Therefore, the assumption—that metallic ions undergo reduction to metallic atoms that aggregate to form the nanoparticles—did not hold for almost all of the common reducing agents. Recent DFT calculations [91,116,117] revealed that the mechanism of metallic nanoparticle formation, especially of AuNPs and AgNPs, consists of several steps that begin with the formation of agglomerates with M^I^-M^I^ (M = Ag or Au) bonds and M^0^-M^I^ bonds that are eventually reduced to form Ag_n_^0^ or Au_n_^0^ as the intermediate to AgNPs or AuNPs. Silver and gold atoms are known, respectively, to form Ag–Ag [118,119] and Au–Au [75,120,121,122,123,124,125] bonds and form clusters. Bond formation is also known to occur between silver atoms and silver ions [92,119,126,127,128], between silver ions [129,130,131], and between gold ions [80,120,131,132]. Henglein et al. [92,119,127] experimentally studied the formation of small silver clusters and small clusters in which silver ions were involved. Jin et al. [120] emphasized the importance of protecting organic ligands, especially thiolate, when attempting to obtain such clusters [124]. The role of thiolate was also studied by Barngrover et al. [62], while Polte et al. [84] demonstrated the role of stabilizing agents. Zhu et al. [80] and Wu et al. [133] found that the formation of the Au^I^SR intermediate is critical to obtain Au_25_ nanoclusters of high purity and in large yields. Loh et al. [132] used ab initio calculations on gold nucleation to suggest that strong gold–gold atom coupling and water-mediated metastable gold complexes are involved in the nucleation of nanocrystals in aqueous solutions. Understanding the intermediate steps in nuclei formation has important implications for the formation and growth of both crystalline and amorphous materials. Zavras et al. [87] used DFT calculations to show that [Ag_3_(H)_2_−x(BH_4_)_x_L_n_]^+^ clusters provide a direct link between mixed hydride/borohydride silver clusters (x = 1), dihydride silver clusters (x = 0) and silver clusters. In DFT-based research, Barngrover et al. [64] showed how Au^0^-containing species can be formed without assuming the formation of Au^0^ atoms (radical species), using methanol and benzene solvents as models for one-phase and two-phase gold nanoparticle growth processes, the precursors of the AuNP were AuCl_2_^−^, AuBr_2_^−^, AuI_2_^−^, AuClPH_3_, and AuCl(H)SCH_3_^(−)^ that initially form Au^0^ clusters: Au_2_ X _2_^2−^ (X = Cl^−^, Br^−^, I^−^) and then Au_3_Cl_3_^2−^. Larger clusters are formed by reactions between these clusters or between such clusters and additional AuCl_2_^−^ species [64]. 

## 5. Formation of AgNPs and AuNPs—A New Approach

Since the simple reduction of Ag^+^(aq) or Au^+^(aq) to form a hydrated Ag^0^_aq_ or Au^0^_aq_ atom, respectively, is not possible with most reduction agents, a new approach entailing a multi-step reaction was proposed [91,116,117]. The initial step in this approach, prior to the reduction, is the formation of clusters in which Ag^+^-Ag^+^ or Au^+^-Au^+^ bonds are involved via the formation of the NPs.

The complexity of AgNP and AuNP formation was demonstrated experimentally [91,117]. In these studies, the precursor of the AgNPs or AuNPs was an aqueous solution of Ag(H_2_O)_2_^+^ or AuCl_2_^−^, and aqueous BH_4_^−^ was used as a reduction agent. Spectroscopic results obtained by using a stop-flow method clearly show that the reduction of Ag(H_2_O)_2_^+^ or AuCl_2_^−^ to form AgNPs or AuNPs, respectively, is a multi-step reaction. At least four [91] or three [117] assembly and disassembly processes were observed in a very short interval of time prior to the formation of AgNPs or AuNPs, respectively. 

DFT calculations have shown that Ag(H_2_O)H is the first intermediate that is formed in the process of AgNP production from Ag(H_2_O)_2_^+^ using H_2_ [116] or BH_4_^−^ [91] in aqueous solutions. AgNPs are formed via the reaction of Ag(H_2_O)H with another silver ion and oligomerization followed by the release of H_2._ The first step of this process is [116]:Ag(H_2_O)H + Ag(H_2_O)_2_^+^ → Ag_2_(H_2_O)_3_H^+^ ΔG^0^ = −19.38 kcal/mol (ΔG^#^ = 2.46 kcal/mol)

After oligomerization, the release of H_2_ and the reduction of Ag^+^ to Ag^0^ is thermodynamically possible, e.g.:(Ag(H_2_O)H)_2_Ag^+^ → (Ag(H_2_O)_2_Ag^+^ + H_2_ ΔG^0^ = −1.99 kcal/mol

When BH_4_^−^(aq) was used as the reducing agent, the formation of Ag(H_2_O)H was just a side reaction, because the silver cations mainly catalyze the hydrolysis of BH_4_^−^. This explains why the Creighton process [134] (BH_4_^−^ is used as the reducing agent) requires such a large excess of BH_4_^−^ to reduce the silver cations.

In contrast to these results, the starting point for the formation of AuNPs [117] using the Creighton process [134] is AuH_2_^−^, as the production of Au(H_2_O)H is unfavored thermodynamically [117]. Au^3+^ is reduced by BH_4_^−^ to Au^+^ in a very exothermic reaction, and this reduction is followed by some oligomerization steps. The spontaneous release of H_2_ is the starting point in the reduction of Au^+^ to Au^0^ via the formation of AuNPs [117]:Au_6_H_8_^2^ˉ → Au_6_H_6_^2^ˉ + H_2_ Δ*G*^0^ = −0.96 kcal/mol

In the Au system, when BH_4_^−^(aq) is used as a reduction agent (Creighton procedure [134]), the reduction of the central cation is the main process [117]. In the silver system, in contrast, the Ag^+^ ions mainly catalyze the BH_4−_ hydrolysis, and the reduction of the central cation is only a side process. In both procedures, BH_4_^−^ ions ligate the central metal cation to form Ag(H_2_O)BH_4_ or AuCl(BH_4_)^−^ and Ag(BH_4_)_2_^−^ or Au(BH_4_)_2_^−^. But while the silver ion hydrolysis of these species forms several Ag(H_2_O)(BH_4-n_OH_n_)^−^, *n* = 1–3 and Ag(BH_4-n_OH_n_)(BH_4-n′_OH_n′_)^−^, *n* = 1–3 and *n*′ = 1–3 intermediates [91], the hydrolysis of the BH_4_^−^ ligands in the gold system only forms BH_3_OH^−^ ligands, and further hydrolysis of the ligands results in the loss of the BH(OH)_2_ intermediate and the formation of the hydrides AuClH^−^ and AuH_2_^−^ [117]. The ΔG^0^ and ΔG^#^ values are very similar in the Au^+^ and Ag^+^ systems for one BH4^−^ ligand or for two such ligands. A comparison of these values is given in Table 2.

The ΔG^#^ values are significantly lower than the value for the first hydrolysis of BH_4_^−^ in absence of the metallic cations, which catalyze the hydrolysis, even though the ΔG^0^ value is quite similar [91]:BH_4_^−^_aq_ + H_2_O → [BH_3_(OH)]^−^_aq_ + H_2_ Δ*G*^0^ = −0.07 kcal/mol (Δ*G*^‡^ = 54.78 kcal/mol)

## 6. Conclusions

Noble Metal NPs can be produced by using a variety of bottom-up and top-down physical, chemical, and biological methods that require the addition of chemical stabilizers and biological materials. In addition, methods have been developed to control the morphology, particle size, and structure of the produced NPs, and thus affect their unique physical and chemical properties.

This review concentrated on the bottom-up methods that do not require a stabilization agent. The standard reduction potentials of M^n+^_aq_/M^0^_aq_ differ significantly from those of M^n+^_aq_/M^0^_solid_. The values of E^0^(M^n+^_aq_/M^0^_aq_) for many typical metallic cations are negative in contrast to the positive values of E^0^(M^n+^_aq_/M^0^_solid_) exhibited by the same cations. As a result, the assumption that metallic NP production begins with the simple reduction of M^n+^_aq_ is not valid for most reducing agents, and as such, M^0^- atoms are not produced. The process of NP formation is more complex and involves many steps.

A novel approach to AuNP and AgNP formation is based on the tendency of Ag^+^ and Au^+^ to agglomerate and form M^I^–M^I^, M^I^–M and M–M bonds. According to this new approach [91,116,117], the reduction is a multi-step process, the initial step in which is the formation of clusters that involve Ag^+^–Ag^+^ or Au^+^–Au^+^ bonds prior to the reduction and release of H_2_ via the formation of the NPs. The potential applicability of this novel approach in the context of the formation of AgNPs, AuNPs, and other metallic NPs warrants its further investigation, both experimentally and theoretically and with other metallic cations and reducing agents.

## Figures and Tables

**Figure 1 molecules-26-02968-f001:**
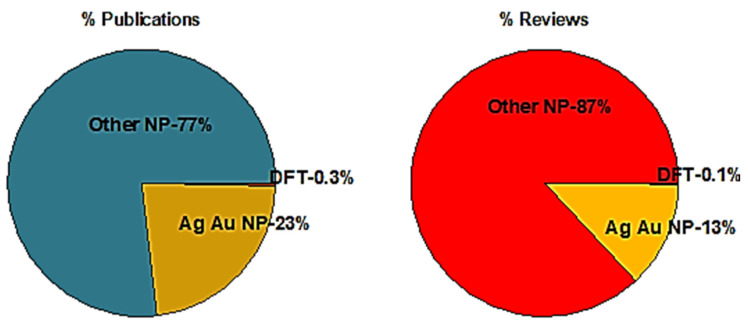
Percentages of publications by subject, 2000–2020, according to SciFinder.

**Figure 2 molecules-26-02968-f002:**
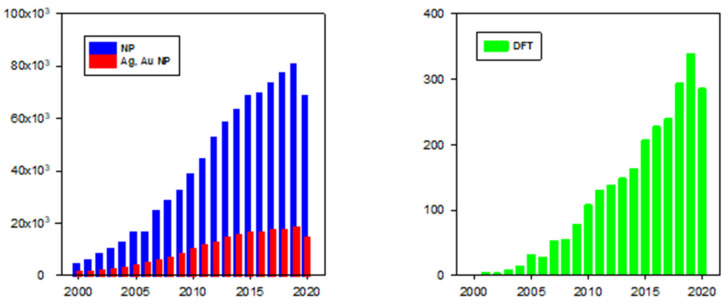
Number of publications in years 2000–2020, results according to SciFinder.

**Table 1 molecules-26-02968-t001:** Standard reduction potentials versus SHE; data from references [89,91].

Metal	Redox Couple	E^0^(M^n+^/) (M^0^_solid_) V vs. SHE [89]	E^0^(M^n+^/) (M^0^_aq_) V vs. SHE [91]
Zinc	Zn^2+^_(aq)_/Zn^0^	−0.76	−1.25
Cadmium	Cd^2+^_(aq)_/Cd^0^	−0.4	−0.8
Mercury	Hg^2+^_(aq)_/Hg^0^	+0.85^a^	0.68
Copper	Cu^+^_(aq)_/Cu^0^	0.52	−2.57
Silver	Ag^+^_(aq)_/Ag^0^	0.8	−1.74
Gold	AuCl_2_^−^_(aq)_/Au^0^ +2Cl^−^_(aq)_	1.15 [114]	−2.23
Nickel	Ni^2+^_(aq)_/Ni^0^	−0.25	−2.24
Palladium	Pd^2+^_(aq)_/Pd^0^	0.95	−0.81
Platinum	Pt^2+^_(aq)_/Pt^0^	1.18	−1.51
Cobalt	Co^2+^_(aq)_/Co^0^	−0.28	−2.25
Rhodium	RhCl_6_^3−^_(aq)_/(Rh^0^+6Cl^−^_(aq)_)	+0.43	−1.33
Iridium	IrCl_6_^2−^_(aq)_/(Ir^0^ + 6Cl^−^_(aq)_)	+0.77	−0.83
Iron	Fe^2+^_(aq)_/Fe^0^	−0.44	−2.36
Ruthenium	RuCl_3(aq)_/(Ru^0^ + 3Cl^−^_(aq)_)	0.51 [115]	−1.55
Osmium	(OsO_4(aq)_ + 8H^+^_(aq)_)/(Os^0^ + 4H_2_O)	+0.84	−0.12
Manganese	Mn^2+^_(aq)_/Mn^0^	−1.19	−2.43
Chromium	Cr^3+^_(aq)_/Cr^0^	−0.74	−1.95
Vanadium	V^2+^_(aq)_/V^0^	−1.17	−5.08
Titanium	Ti^3+^_(aq)_/Ti^0^	−1.37	−3.17

**Table 2 molecules-26-02968-t002:** ΔG^0^ and ΔG^#^ values for the first hydrolysis of BH_4_^−^.

Reaction	ΔG^0^ (kcal/mol)		ΔG^#^ (kcal/mol)	
	M = Ag^+^ [91]	M = Au^+^ [117]	M = Ag^+^ [91]	M = Au^+^ [117]
ML(BH_4_)^n^ + H_2_O → ML(BH_3_OH)^n^ + H_2_ ^a^	0.38	−3.61	20.09	23.73
M(BH_4_)_2_^−^ + H_2_O → M(BH_4_)(BH_3_OH)^−^ + H_2_	−0.05	−3.01	19.88	22.16
M(BH_4_)(BH_3_OH)^−^ + H_2_O → M(BH_4_)_2_^−^ + H_2_	0.09	−3.00	19.72	20.39

^a^ For M = Ag^+^ L = H_2_O and *n* = 0, for M = Au^+^ L = Cl^−^ and *n* = −1.

## Data Availability

Not applicable.

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
