# Peer review of "Mechanism of Producing Metallic Nanoparticles, with an Emphasis on Silver and Gold Nanoparticles, Using Bottom-Up Methods"

_molecules, 2021, doi:10.3390/molecules26102968_

Round 1

Reviewer 1 Report

The revised version is recommended for publication.

Author Response

Thank you for recommendation:

The revised version is recommended for publication.

Reviewer 2 Report

Overall, the work is reasonable (on a scale of 1 to 10: 6).   

I would like to invite the authors to carefully read the entire text again, and see if at numerous places the story could be made somewhat more fluent. There are sections which simply consist of a bunch of sentences referring to literature, all starting in the same style (X et al ...... {place a comment on data / conclusions / reactions from work of X} + [reference to work of X]) and that is somewhat disappointing. Such a style doesn't read smooth at all. When the authors aspire to write a Review, their overall style and 'story telling' can definitely be even further improved. 

Minor comments: 

Line 31: Plasmonic Properties > Plasmonic properties

Are Figure 1 and 2 really necessary to include ? It is nice to see a field develop, however, this is not really adding too much in the understanding. 

Line 61 + 66 both contain the caption of Figure 1. 

Line 93: that that 

Line 200: guess the '25' should be in subscript for: Au25

Line 253: BH4- > check the correct sub/superscript

Line 260: Nobel Metal > Noble metal (mind the difference between 'nobel' and 'noble')

Check for consistency with the style in the rest of the work the column headers of Table 1. Is it Mn or Mn+

Line 267: is this consistent with the rest: [ Ms ] ? Column header of Table 1 uses: [ M0solid ]

Line 272: perhaps change [ the reduction process is a multi-step procedure. ] > [ the reduction is a multi-step process. ] 

Author Response

9 May 2021

Dear Prof. Kayla Li,

 Manuscript ID (molecules-1125858):

Thank you and the reviewers for evaluating the manuscript again.

Following the request of reviewer 2:

" I would like to invite the authors to carefully read the entire text again, and see if at numerous places the story could be made somewhat more fluent. There are sections which simply consist of a bunch of sentences referring to literature, all starting in the same style (X et al ...... {place a comment on data / conclusions / reactions from work of X} + [reference to work of X]) and that is somewhat disappointing. Such a style doesn't read smooth at all. When the authors aspire to write a Review, their overall style and 'story telling' can definitely be even further improved." 

The manuscript was sent for linguistic editing, and we hope that now you will find this manuscript suitable for publication in “Molecules”.

We also corrected all minor comments: 

"Line 31: Plasmonic Properties > Plasmonic properties

Are Figure 1 and 2 really necessary to include ? It is nice to see a field develop, however, this is not really adding too much in the understanding. 

Line 61 + 66 both contain the caption of Figure 1. 

Line 93: that that 

Line 200: guess the '25' should be in subscript for: Au25

Line 253: BH4- > check the correct sub/superscript

Line 260: Nobel Metal > Noble metal (mind the difference between 'nobel' and 'noble')

Check for consistency with the style in the rest of the work the column headers of Table 1. Is it Mn or Mn+ ? 

Line 267: is this consistent with the rest: [ Ms ] ? Column header of Table 1 uses: [ M0solid ]

Line 272: perhaps change [ the reduction process is a multi-step procedure. ] > [ the reduction is a multi-step process. ]" 

Sincerely,

 Haya Kornweitz

Round 2

Reviewer 2 Report

The authors have taken into consideration the recommendation to have a critical reading done of the work, which has improved the overall readability. All minor suggestions / improvements have been taken into consideration. Therefore, the work can be accepted in its current form. 

Few minor comments: 

[1]

Line 226-228

[This explains why the Creighton process [134] (BH4- is used as the reducing agent) requires such a large excess of BH4- to reduce the silver cations.]

The part in between ( ... ) is duplicating what is mentioned in the main sentence. 

Perhaps change into: 

[This explains why in the Creighton process [134] such a large excess of reducing agent (BH4-) is required to reduce the silver cations.]    

[2]

Line 251

[Noble Metal] > [Noble metal]

This manuscript is a resubmission of an earlier submission. The following is a list of the peer review reports and author responses from that submission.

Round 1

Reviewer 1 Report

The review manuscript is well organized and the mechanism of producing Au and Ag nanoparticles was discussed in details. It is publishable in Molecules in my opinion.

Minor comment: There is no data in Figure 2 (above).

Author Response

Minor comment: There is no data in Figure 2 (above).

Answer: The figures were corrupted in the pdf version. I inserted them now as jpeg, and it seems as the problem is solved.

Reviewer 2 Report

Overall, the work is reasonably readable, although sometimes some more explaining and clarifying comments by the authors would help, instead of referring to the work in literature, from which a few conclusions and insights are copied and sometimes postulated without further clarification and attempts to link these together. 

I would have a few questions and suggestions, listed below: 

========================  

Line 13, 14, 18: carefully check for 'n' to be consistently in superscript 

Line 25: no need to put 'Silver' with capital 'S' 

Line 34-35: 'reviewed ....... reviews' 

Figure 1: when 'Reviews' is written with capital 'R' then 'publications' should be the same style. Perhaps use 'Publications' (as multiple was used in figure caption)

Figure 2(a) shows an empty graph without any bars. '.' missing at the end of the caption

Line 62: 'AgNp' should be 'AgNP'

Line 63: should the '9' be in subscript ?

Line 84: for reference [41] the full names of the authors are mentioned, this is not in line with the style used previously in the Review.

Line 96: the '4' of the borohydride should be in subscript

Line 96-102: 'The positive reduction potential of AuIII and AuI given above and of Ag+/Ag (0.799 V have convinced the scientific community that the first step of formation of AuNP and AgNP is simply a reduction process in which gold or silver atoms are formed.' > HOW does the value of positive redox potentials convince the scientific community ? That should properly be explained and not just mentioned. To me this is unclear why that is the case ! To put it very clear, when the redox potential of ANY reducing agent is negative WITH RESPECT TO THE VALUE of the redox potential for the metal ion reduction reaction, one might expect a reducing agent to be spontaneously reacting with that metal ion species in solution. In my opinion, it suffices to discuss the RELATIVE positions of the redox potentials of reducing agent and metal ion (by taking a look at the tabulated E0 values), in order to decide if thermodynamics would allow for a SPONTANEOUS reaction.  To me this mentioning of POSITIVE would only make sense and be correct when one is using hydrogen as the reducing agent. This part should properly be explained in the work, especially given the fact this is the crux of the authors' work ! 

Is there a specific reason why different characters are used in the redox potentials ? '0' and

'º' are used in the text and in Table 1 ? For instance on line 113 + 114 both are used and it is not immediately clear to me why that is the case. 

Table 1, last column: a number of values (-2.57, -2.23, -2.25, - 2.36, -1.55, -5.08, -3.17 V vs SHE) are well outside the water stability window. How should we interpret these values ? What are these values physically / chemically implying ? Are these species then stable or not ? This would deserve a proper explanation by the authors.  

Line 124-125: 'The assumption that the metallic ions are reduced to form metallic atoms and the aggregation 124 of these atoms forms the nanoparticles failed for almost all the common reducing agents.' > This statement falls out of the sky, especially at the begin of a new section without any context given, yet. What is the strong conclusion based on ? 

Line 166: is 'ΔG#0 = 2.46 kcal/mol' a typo ? Should that be 'ΔG# = 2.46 kcal/mol' ?

Line 189: the '4' in the borohydride formula should be in subscript

Line 197-198: 'E0(Maqn+/M0aq) values for many typical metallic cations are negative in contrast to the positive 197 E0(Maqn+/Ms) values for the same cations' > Similar comment as my previous one about negative / positive.  

========================

Author Response

Reviewer 2

Overall, the work is reasonably readable, although sometimes some more explaining and clarifying comments by the authors would help, instead of referring to the work in literature, from which a few conclusions and insights are copied and sometimes postulated without further clarification and attempts to link these together. 

I would have a few questions and suggestions, listed below: 

========================  

Line 13, 14, 18: carefully check for 'n' to be consistently in superscript 

Answer: I correct all along the text and now it is always written: E0(Mn+aq/M0solid) E0( Mn+aq/M0aq

Line 25: no need to put 'Silver' with capital 'S' 

Answer: corrected

Line 34-35: 'reviewed ....... reviews' 

Answer: It is changed to: reviewed in numerous articles

Figure 1: when 'Reviews' is written with capital 'R' then 'publications' should be the same style. Perhaps use 'Publications' (as multiple was used in figure caption)

Answer: corrected

Figure 2(a) shows an empty graph without any bars. '.' missing at the end of the caption

Answer: The figures were corrupted in the pdf version. I inserted them now as jpeg, and it seems as the problem is solved

Line 62: 'AgNp' should be 'AgNP'

Answer: corrected. Thank you.

Line 63: should the '9' be in subscript ?

Answer:

Yes. We add for clarification: (a cluster of nine atoms)

Line 84: for reference [41] the full names of the authors are mentioned, this is not in line with the style used previously in the Review.

Answer: corrected, thank you.

Line 96: the '4' of the borohydride should be in subscript

Answer: corrected, thank you.

Line 96-102: 'The positive reduction potential of AuIII and AuI given above and of Ag+/Ag (0.799 V have convinced the scientific community that the first step of formation of AuNP and AgNP is simply a reduction process in which gold or silver atoms are formed.' > HOW does the value of positive redox potentials convince the scientific community ? That should properly be explained and not just mentioned. To me this is unclear why that is the case ! To put it very clear, when the redox potential of ANY reducing agent is negative WITH RESPECT TO THE VALUE of the redox potential for the metal ion reduction reaction, one might expect a reducing agent to be spontaneously reacting with that metal ion species in solution. In my opinion, it suffices to discuss the RELATIVE positions of the redox potentials of reducing agent and metal ion (by taking a look at the tabulated E0 values), in order to decide if thermodynamics would allow for a SPONTANEOUS reaction.  To me this mentioning of POSITIVE would only make sense and be correct when one is using hydrogen as the reducing agent. This part should properly be explained in the work, especially given the fact this is the crux of the authors' work ! 

Answer:

Yes, this is really our main point. To clarify, we add:

To get a spontaneous reaction a positive potential is required. The given positive reduction potentials of the metallic ions are potentials of a half reaction (reduction) but as they are positive, for any common reducing agent, the redox potential will be positive, and the reduction will be spontaneous.

Is there a specific reason why different characters are used in the redox potentials ? '0' and

'º' are used in the text and in Table 1 ? For instance on line 113 + 114 both are used and it is not immediately clear to me why that is the case. 

Answer: corrected, thank you, E0  is now written everywhere in table 1.

Table 1, last column: a number of values (-2.57, -2.23, -2.25, - 2.36, -1.55, -5.08, -3.17 V vs SHE) are well outside the water stability window. How should we interpret these values ? What are these values physically / chemically implying ? Are these species then stable or not ? This would deserve a proper explanation by the authors.  

Answer: To clarify we add:

The low values of E0(Mn+aq/M0aq) in this table indicate that it is impossible to get solvated zero-valent atoms in an aqueous solution, only reduction to a metallic solid is possible.

Line 124-125: 'The assumption that the metallic ions are reduced to form metallic atoms and the aggregation 124 of these atoms forms the nanoparticles failed for almost all the common reducing agents.' > This statement falls out of the sky, especially at the begin of a new section without any context given, yet. What is the strong conclusion based on ? 

Answer: To clarify we add:

As the redox potential E0(Mn/(M0aq) is significantly lower than the values of E0(Mn/(M0solid)  for all the metals, and these values are negative for many common metallic ions, the assumption that the metallic ions are reduced to form metallic atoms and the aggregation of these atoms forms the nanoparticles failed for almost all the common reducing agents

Line 166: is 'ΔG#0 = 2.46 kcal/mol' a typo ? Should that be 'ΔG# = 2.46 kcal/mol' ?

Answer: corrected, thank you.

Line 189: the '4' in the borohydride formula should be in subscript

Answer: corrected, thank you.

Line 197-198: 'E0(Maqn+/M0aq) values for many typical metallic cations are negative in contrast to the positive 197 E0(Maqn+/Ms) values for the same cations' > Similar comment as my previous one about negative / positive

Answer: corrected as before

Reviewer 3 Report

This review manuscript describes an overview of the mechanism of fabrication of metallic nanoparticles, focusing on the Ag and Au NPs. The authors compare the mechanism in which the nanoparticles are formed after the reduction of the metal salt into metallic clusters and consequently their aggregation forms the metal nanoparticles (mechanism used nowadays) with a new approach in which the formation of metallic NPs is more complex and involve intermediates.

The manuscript is well structured however there are some points that should be revised.

I do not recommend the manuscript for publication after major revisions.

  1. The new approach proposed in reference 75, 100 and 101 is for spherical nanoparticles or all morphologies and shapes? We already know from the literature that in Au nanorods, the gold salt (Au3+) reduces first to Au+ and then to Au0. Can the authors comment?
  2. Figure 1 should have different colors for each point. For instance, in the top figure the other NPs, Ag Au NPs and DFT should have 3 different colors
  3. Figure 2, first graph does not should anything
  4. Page 4, line 63: it is Au9 and Ag9 or it is Au0 and Ag0?
  5. In my opinion if the authors write some reduction agents by their name they should write all (page 5, lines 117-118)
  6. In table 1 all the zero (0) is written o. Please revised
  7. It is the last formula (last column) in table 1 correct?
  8. What is SHE (line 111)? Or nF (line 108)? There are lot of abreviations in the mani text and formulas that are not described. Please revised or add a abbreviation column at the end of the manuscript

Author Response

Reviewer 3

This review manuscript describes an overview of the mechanism of fabrication of metallic nanoparticles, focusing on the Ag and Au NPs. The authors compare the mechanism in which the nanoparticles are formed after the reduction of the metal salt into metallic clusters and consequently their aggregation forms the metal nanoparticles (mechanism used nowadays) with a new approach in which the formation of metallic NPs is more complex and involve intermediates.

The manuscript is well structured however there are some points that should be revised.

I do not recommend the manuscript for publication after major revisions.

  1. The new approach proposed in reference 75, 100 and 101 is for spherical nanoparticles or all morphologies and shapes? We already know from the literature that in Au nanorods, the gold salt (Au3+) reduces first to Au+ and then to Au0. Can the authors comment?

Answer:

This manuscript deals with nascent nanoparticles, the first steps leading from ions to small clusters that aggregate to form nanoparticles, it doesn’t refer to morphology and shapes.

We add (lines 75-76):

We refer only to the early first steps that lead from aqueous ions to small metallic clusters that aggregate to form nanoparticles, it doesn’t refer to morphology and shapes.

  1. Figure 1 should have different colors for each point. For instance, in the top figure the other NPs, Ag Au NPs and DFT should have 3 different colors

Answer: The figures were corrupted in the pdf version. I inserted them now as jpeg, and it seems as the problem is solved.

  1. Figure 2, first graph does not should anything

Answer: The figures were corrupted in the pdf version. I inserted them now as jpeg, and it seems as the problem is solved.

  1. Page 4, line 63: it is Au9 and Ag9 or it is Au0 and Ag0?

Answer:

Yes. We add for clarification: (a cluster of nine atoms)

  1. In my opinion if the authors write some reduction agents by their name they should write all (page 5, lines 117-118)

Answer: corrected, thank you.

  1. In table 1 all the zero (0) is written o. Please revised

Answer: corrected, thank you.

  1. It is the last formula (last column) in table 1 correct?

Answer: Thank you, it is corrected to: E0(Mn/(M0aq) V vs. SHE

  1. What is SHE (line 111)? Or nF (line 108)? There are lot of abreviations in the mani text and formulas that are not described. Please revised or add a abbreviation column at the end of the manuscript

Answer: We add an explanation:

(n-number of electrons involved in the redox reaction, F-faraday constant)

Round 2

Reviewer 2 Report

I would like to thank the authors for taking into consideration the feedback and applying the suggested changes as well as the attempts to improve the work. 

=====  

Regarding below addition of text:

To get a spontaneous reaction a positive potential is required. The given positive reduction
potentials of the metallic ions are potentials of a half reaction (reduction) but as they are
positive, for any common reducing agent, the redox potential will be positive, and the
reduction will be spontaneous.  

In my opinion this addition is not improving the readability and level of explaining.

[ To get a spontaneous reaction a positive potential is required.] > This is suggesting that metal ions with a negative potential cannot be reduced by a spontaneous reaction with a reducing agent. How do the authors then explain the existence of electroless deposition baths that are yielding metallic cobalt and nickel ? 

[ for any common reducing agent ] > This is a vague formulation ! 'common' is very subjective. Which reducing agents are the authors referring to ? Perhaps also good to include the standard potential of the reducing agent, so that it can be compared to the metal ion that can / cannot be reduced spontaneously by it. This comparison of 2 potentials can then be used to explain properly what the authors are failing to do with their words.  The text from line 129-133 might better be of use at this position in the work. 

The total count of 'common' in the work is 8. Perhaps assess if that is needed. It would be good to avoid 'common' in front of 'reducing agent'. 

[ as they are positive, for any common reducing agent, the redox potential will be positive, ] > This is an extremely confusing way of formulation. In the added sentence the word 'positive' is used three times and to me it is still unclear what the message is. It can easily be misread and interpreted in a wrong way. 

It would also be good to properly define / explain what the authors mean by [ reduction potential ] and [ redox potential ]. It is very common (pun intended) to use them interchangeably, however, the authors seem to reserve different meanings for them in their added text. Using both of them in the same sentence is, perhaps, not the best approach to clarify things. 

=====  

Regarding below addition of text:

The low values of E0(Mn+aq/M0aq) in this table indicate that it is impossible to get solvated
zero-valent atoms in an aqueous solution, only reduction to a metallic solid is possible.

It is not explained why that is impossible, it is just mentioned that it is the case. Stating facts is not explaining things, certainly not a way to formulate in a Review. 

=====   

Line 14/15 , Abstract

[ Therefore, many common moderate reduction agents cannot reduce the metallic cations to zero-valent atoms, ] > See my previous remarks about 'common'. Also 'moderate' is vague when not used in the proper relation to what the subject is being compared to, or without quantifying the standard potential of the reducing agent the authors are referring to. 

=====   

Line 80: 

The button-up procedure > The bottom-up procedure

Reviewer 3 Report

The manuscript can be accepted in it present form for publication in Molecules